# The alignment property of SGD noise and how it helps select flat minima: A stability analysis

**Lei Wu**
School of Mathematical Sciences
Peking University

**Mingze Wang**
School of Mathematical Sciences
Peking University

**Weijie J. Su**
Department of Statistics and Data Science
University of Pennsylvania

## Abstract

The phenomenon that stochastic gradient descent (SGD) favors flat minima has played a critical role in understanding the implicit regularization of SGD. In this paper, we provide an explanation of this striking phenomenon by relating the particular noise structure of SGD to its *linear stability* (Wu et al., 2018). Specifically, we consider training over-parameterized models with square loss. We prove that if a global minimum $\theta^*$ is linearly stable for SGD, then it must satisfy $\|H(\theta^*)\|_F \leqslant O(\sqrt{B}/\eta)$, where $\|H(\theta^*)\|_F, B, \eta$ denote the Frobenius norm of Hessian at $\theta^*$, batch size, and learning rate, respectively. Otherwise, SGD will escape from that minimum *exponentially* fast. Hence, for minima accessible to SGD, the sharpness—as measured by the Frobenius norm of the Hessian—is bounded *independently* of the model size and sample size. The key to obtaining these results is exploiting the particular structure of SGD noise: The noise concentrates in sharp directions of local landscape and the magnitude is proportional to loss value. This alignment property of SGD noise provably holds for linear networks and random feature models (RFMs), and is empirically verified for nonlinear networks. Moreover, the validity and practical relevance of our theoretical findings are also justified by extensive experiments on CIFAR-10 dataset.

## 1 Introduction

Modern machine learning (ML) models are often operated with far more unknown parameters than training examples, a regime referred to as over-parameterization. In this regime, there are many global minima, all of which have zero training loss but their test performance can be significantly different [47]. Fortunately, it is often observed that SGD converges to those generalizable ones, even without needing any explicit regularizations [49]. This suggests there must exist certain "implicit regularization" mechanism at work [32, 14, 47, 6].

More mysteriously, SGD solutions often generalize better than gradient descent (GD) solutions [21, 38]. Therefore, the SGD noise must play a critical role in implicit regularization. The most popular explanation is that SGD favors flatter minima [21] and flatter minima generalize better [16]. This flat-minima principle has been extensively and successfully adopted in practice to tune the hyperparameters of SGD [44, 21] and to design new optimizers [17, 11, 43] for improving generalization. Therefore, understanding how SGD noise biases SGD towards flatter minima is of paramount importance, which is the main focus of this paper.

The works [44, 10, 48] show that SGD noise is highly anisotropic; [30, 41] find the magnitude of SGD noise to be loss dependent. Both structures are shown to be critical for SGD picking flat minima.

36th Conference on Neural Information Processing Systems (NeurIPS 2022).

However, these works all make unrealistic (even wrong) over-simplifications of SGD noise (see the related work section for more details) in their analysis. In addition, instead of studying SGD, they all consider the continuous-time stochastic differential equation (SDE), which is a good modeling of SGD only in finite time and small learning rate (LR) regime [25]. It is generally unclear how this SDE modeling is relevant for understanding SGD with a large LR—a regime preferred in practice. Consequently, these works only provide intuitive and empirical analyses, lacking a quantitative characterization of when and how SGD favors flat minima.

Another line of works [46, 29] relate the selection bias of SGD to the *dynamical stability*. In over-parameterized case, all global minima are fixed points of SGD but their dynamical stabilities can be very different. At unstable minima, a small perturbation will drive SGD to leave away, whereas, for stable minima, SGD can stay around and even converge back after initial perturbations. Thus SGD prefers stable minima over unstable ones. Specifically, [46, 29] analyze the linear stability [1] of SGD, showing that a linearly stable minimum must be flat and uniform. Different from SDE-based analysis, this stability-based analysis is relevant for large-LR SGD and is even empirically accurate in predicting the properties of minima selected by SGD [46, 20, 8].

In this work, we follow the linear stability analysis in [46, 29] but take the particular structure of SGD noise into consideration. We establish a direct connection between linear stability and flatness, which allows us to obtain a quantitative characterization of how the learning rate and batch size affect the flatness of minima accessible to SGD. In contrast, [46, 29] have to introduce another quantity: the non-uniformity together with flatness to characterize linear stability because of neglecting the noise structure.

**Setup** Let $\{(x_i, y_i)\}_{i=1}^n$ with $x_i \in \mathbb{R}^d, y_i \in \mathbb{R}^K$ be the training set and $f(\cdot; \theta)$ with $\theta \in \mathbb{R}^p$ be our model. The *model size* is defined to be the number of parameters $p$ and in this paper. For simplicity, we will always assume $K = 1$ and the extension to the case of $K > 1$ is straightforward. Let $L_i(\theta) = \frac{1}{2}|f(x_i; \theta) - y_i|^2$ be the fitting error at the $i$-th sample and $L(\theta) = \frac{1}{n}\sum_{i=1}^n L_i(\theta)$ be the empirical risk. We shall focus on the over-parameterized case in the sense that $\inf_\theta L(\theta) = 0$. To minimize $L(\cdot)$, we consider the mini-batch SGD:

$$\theta_{t+1} = \theta_t - \frac{\eta}{B}\sum_{i \in I_t} \nabla L_i(\theta_t), \tag{1}$$

where $\eta$ and $B$ are the learning rate and batch size, respectively. This SGD can be rewritten as $\theta_{t+1} = \theta_t - \eta(\nabla L(\theta_t) + \xi_t)$, where $\xi_t$ is the noise, satisfying $\mathbb{E}[\xi_t] = 0$ and $\mathbb{E}[\xi_t \xi_t^T] = \Sigma(\theta_t)/B$. Here the noise covariance $\Sigma(\theta) = \frac{1}{n}\sum_{i=1}^n \nabla L_i(\theta)\nabla L_i(\theta)^T - \nabla L(\theta)L(\theta)^T$. To characterize the local geometry of loss landscape, we consider the Fisher matrix: $G(\theta) = \frac{1}{n}\sum_{i=1}^n \nabla f(x_i; \theta)\nabla f(x_i; \theta)^T$ and the Hessian matrix: $H(\theta) = G(\theta) + \frac{1}{n}\sum_{i=1}^n (f(x_i; \theta) - y_i)\nabla^2 f(x_i; \theta)$. When the loss value is small, $H(\theta) \approx G(\theta)$ and in particular, if $\theta^*$ is an global minimum, $H(\theta^*) = G(\theta^*)$.

**Notations.** For a vector $a$, let $\|a\| = \sqrt{a^T a}$ and $\|a\|_W = \sqrt{a^T W a}$. For a matrix $A$, denote by $\{\lambda_j(A)\}$ the eigenvalue of $A$ in a decreasing order. For other notations, we refer to Appendix C.

**Our main contributions** are summarized as follows.

- We first show that for many ML models, the SGD noise is geometry aware: 1) the noise magnitude is proportional to the loss value; 2) the noise covariance aligns well with the Fisher matrix. Specifically, to quantify the alignment strength, we define a loss-scaled alignment factor $\mu(\theta)$, which is proved to be bounded from below, i.e., there exists a *size-independent* positive constant $\mu_0$ such that $\mu(\theta) \geqslant \mu_0$, for linear networks (Proposition 2.3) and RFMs (Proposition 2.5), and is also empirically justified for nonlinear networks. Moreover, we identify that it is the uniformity of model gradient norms $\{\|\nabla f(x_i; \theta)\|_{G(\theta)}\}_i$ that accounts for this *alignment property* of SGD noise.

- We then provide a thorough analysis of the linear stability of SGD by exploiting the alignment property of noise. We prove in Theorem 3.3 that if a global minimum $\theta^*$ is linearly stable, then $\|H(\theta^*)\|_F \leqslant \eta^{-1}\sqrt{B/\mu_0}$. Here the constant $\mu_0$ quantifies the alignment strength of SGD noise. Hence, for minima accessible to SGD, the Hessian's Frobenius norm—the flatness perceived by SGD—is bounded independently of the model size and sample size. Moreover, if a minimum is too sharp, violating the preceding stability condition, SGD will escape from it *exponentially* fast (Theorem 3.5). Together, we obtain a quantitative characterization of when and how much SGD dislikes sharp minima.

- Our theoretical findings are also corroborated with well-designed experiments on a variety of models including linear networks, RFMs, convolutional networks, and fully-connected networks. In particular, the practical relevance is demonstrated in Section 4 by extensive experiments on classifying full CIFAR-10 dataset with VGG nets and ResNets.

## 1.1 Related work

**Noise structures.** [52, 19, 27] consider the Hessian-based approximation: $\Sigma(\theta) \approx \sigma^2 H(\theta)$, where $\sigma$ is a small constant. [53] proposes an improved version: $\Sigma(\theta) \approx 2L(\theta)H(\theta)$. But these approximations in general cannot be accurate since Hessian is not semi-positive definite (SPD) in non-convex region. More recently, [30] and [41] study SGD by assuming $\Sigma(\theta) = 2L(\theta)H(\theta^*)$, where $\theta^*$ is a minimum of interest, and $\Sigma(\theta) = \sigma^2 L(\theta)I_p$, respectively. These assumptions completely ignore the state-dependence of noise direction. In contrast, we assume $\Sigma(\theta) = 2L(\theta)C(\theta)$ with $C(\theta)$ having a nontrivial alignment with the Fisher matrix $G(\theta)$, which does not impose any explicit structural assumption on $C(\theta)$. As a result, our assumption is much weaker and can be rigorously justified for popular ML models both theoretically and empirically. More importantly, we show that this weak alignment property is sufficient for analyzing the linear stability of SGD. We anticipate that our alignment assumption can be also adopted to analyze other properties of SGD.

**Escape from sharp minima.** The escape behavior of SGD was first studied in [52, 46], as an indicator of how much SGD dislikes sharp minima. One of the most mysterious observation is that the escape happens in an unreasonably efficient way. However, the theoretical analysis there assumes the noise to be state-independent, and consequently, the derived escape time depends polynomially on the loss barrier. Later [48, 30] attempt to study this issue using the classical diffusion-based framework [12] (Itô-SDE), which cannot explain the unreasonable escape efficiency at all since the resulting escape rates depend on the loss barrier exponentially. See also [30] for an improved analysis. [37, 51] argues that the SGD noise is heavy-tailed and thus SGD should be modeled as Lévy-SDE instead of Itô-SDE. Moreover, it is shown that the heavy-tailedness can ensure the escape rate depends on the basin volume instead of the loss barrier. Unfortunately, the volume in high dimensions always scales with the dimension exponentially and consequently, this does not explain the escape efficiency in high dimensions. Moreover, whether SGD noise is really heavy-tailed and whether the heavy-tailedness is really important for generalization are still debatable for neural networks [40, 26]. In contrast, we show that the unreasonable escape efficiency comes from the particular geometry-aware structure of SGD noise, regardless of whether the noise is heavy- or light-tailed.

**Flatness metrics** In the literature, a variety of flatness metrics have been adopted, such as the largest eigenvalue of Hessian [21], the trace of Hessian [9, 6], the basin volume [51], and the ones scaled by parameter norms [28, 39] in order to achieve the scaling-invariance for ReLU nets. These metrics are proposed for either computation easiness or bounding generalization gaps. It is unclear if they are perceivable to SGD, let alone how the boundedness of them depends on the batch size, learning rate, as well as the model size and sample size. We show that for SGD solutions, the Frobenius norm of Hessian—a flatness perceived by SGD through the linear stability—is bounded by a size-independent quantity. Note that a similar stability argument also applies to GD but only yielding the boundedness of the largest eigenvalue of Hessian [46, 31].

Lastly, we particularly mention the work [33], which provides a fine-grained analysis of the implicit bias of training two-layer diagonal linear networks. This work is related to ours since we both consider the magnitude and direction structure of SGD noise simultaneously. However, the analysis in [33] is limited to the specific toy model but ours is relevant for general models.

## 2 The alignment property of SGD noise

Since $\nabla L_i(\theta) = (f(x_i; \theta) - y_i)\nabla f(x_i; \theta)$, we have the following intuitive approximation [30]:

$$\Sigma(\theta) = \frac{2}{n}\sum_{i=1}^{n} L_i(\theta)\nabla f(x_i;\theta)\nabla f(x_i;\theta)^T - \nabla L(\theta)\nabla L(\theta)^T \overset{(i)}{\approx} \frac{2}{n}\sum_{i=1}^{n} L_i(\theta)\nabla f(x_i;\theta)\nabla f(x_i;\theta)^T$$

$$\overset{(ii)}{\approx} 2\Big(\frac{1}{n}\sum_{i=1}^{n} L_i(\theta)\Big)\frac{1}{n}\sum_{i=1}^{n}\nabla f(x_i;\theta)\nabla f(x_i;\theta)^T = 2L(\theta)G(\theta), \tag{2}$$

where $(i)$ assumes that the full-batch gradient $\nabla L(\theta)$ to be negligible compared with the sample gradients $\{\nabla L_i(\theta)\}$; $(ii)$ assumes that $\{\nabla f(x_i; \theta)\}_i$ and $\{L_i(\theta)\}_i$ are nearly decoupled. This approximation cannot be true in general but tells us that 1) The noise magnitude is proportional to the loss value; 2) the noise covariance aligns with the Fisher matrix.

Motivated by (2), we define $\alpha(\theta) = \frac{\mathrm{Tr}(G(\theta)\Sigma(\theta))}{\|G(\theta)\|_F \|\Sigma(\theta)\|_F}$, $\beta(\theta) = \frac{\|\Sigma(\theta)\|_F}{2L(\theta)\|G(\theta)\|_F}$ . Here $\alpha(\theta)$ quantifies the similarity between $\Sigma(\theta)$ and $G(\theta)$, characterizing how much the noise concentrates in sharp directions of local landscape. $\beta(\theta)$ characterizes the relative non-degeneracy of noise (with respect to the loss value). Then we define the loss-scaled alignment factor:

$$\mu(\theta) = \alpha(\theta)\beta(\theta) = \frac{\mathrm{Tr}(\Sigma(\theta)G(\theta))}{2L(\theta)\|G(\theta)\|_F^2}, \tag{3}$$

which characterizes the direction and magnitude alignment simultaneously. Intuitively speaking, if $\mu(\theta)$ is bounded below, SGD noise is non-degenerate in sharp directions of local landscape. In particular, $\alpha(\theta) = \beta(\theta) = \mu(\theta) = 1$ if the approximation (2) holds.

We say the noise satisfies the $\mu$-**alignment** if $\mu(\theta) > 0$. Compared with the decoupling approximation (2), the $\mu$-alignment is a much weaker condition. Note that this specific weak quantification of alignment is inspired for analyzing the linear stability of SGD, which is the focus of this paper. Specifically, Theorem 3.3 shows that $\mu(\theta)$ along with the Frobenius norm of Hessian determines the linear stability of SGD. One may define other metrics to quantify alignment strength for studying other properties of SGD, but this is beyond the scope of this paper.

**A relaxed alignment.** Let $\Sigma_1(\theta) = \frac{1}{n}\sum_{i=1}^n \nabla L_i(\theta)\nabla L_i(\theta)^T$, $\Sigma_2(\theta) = \nabla L(\theta)\nabla L(\theta)^T$. Then $\Sigma(\theta) = \Sigma_1(\theta) - \Sigma_2(\theta)$. It is often believed that the full-batch gradient $\nabla L$ is relatively small compared to the sample gradients $\{\nabla L_i\}_i$. As a result, the influence of $\Sigma_2(\theta)$ should be negligible compared to $\Sigma_1(\theta)$. To disentangle the influences of them, we define

$$\mu_1(\theta) = \frac{\mathrm{Tr}(\Sigma_1(\theta)G(\theta))}{2L(\theta)\|G(\theta)\|_F^2}, \quad \mu_2(\theta) = \frac{\mathrm{Tr}(\Sigma_2(\theta)G(\theta))}{2L(\theta)\|G(\theta)\|_F^2}.$$

Then $\mu(\theta) = \mu_1(\theta) - \mu_2(\theta)$. Our linear stability analysis in Section 3 show that $\mu_1(\theta) \geqslant \mu_1 > 0$, a condition we refer to as $\mu_1$-**alignment**, is sufficient to ensure that SGD only selects flat minima.

## 2.1 Why does the alignment property hold?

**Definition 2.1** (Norm uniformity of model gradients). *Let* $g_i(\theta) = \nabla f(x_i; \theta)$, $\chi_i(\theta) := \|g_i(\theta)\|_{G(\theta)}^2 = g_i^T(\theta)G(\theta)g_i(\theta)$, $\bar{\chi}(\theta) = \frac{1}{n}\sum_{i=1}^n \chi_i(\theta)$. *Define* $\gamma(\theta) := \min_{i\in[n]} \frac{\chi_i(\theta)}{\bar{\chi}(\theta)}$.

The quantity $\gamma(\theta)$ measures the uniformity of model gradient norms and this property can guarantee the $\mu_1$-alignment as stated below.

**Lemma 2.2.** $\mu_1(\theta) \geqslant \gamma(\theta)$

*Proof.* Noticing $\bar{\chi}(\theta) = \frac{1}{n}\sum_{i=1}^n g_i(\theta)^T G(\theta)g_i(\theta) = \mathrm{Tr}\left(G(\theta)\frac{1}{n}\sum_{i=1}^n g_i(\theta))g_i(\theta)^T\right) = \|G(\theta)\|_F^2$, we have

$$\mathrm{Tr}(\Sigma_1(\theta)G(\theta)) = \frac{2}{n}\sum_{i=1}^n L_i(\theta)\,\mathrm{Tr}(g_i(\theta)g_i(\theta)^T G(\theta))$$

$$= \frac{2}{n}\sum_{i=1}^n L_i(\theta)\chi_i \geqslant \frac{2}{n}\sum_{i=1}^n L_i(\theta)\gamma\bar{\chi} = 2\gamma L(\theta)\|G(\theta)\|_F^2$$

Thus $\mu_1(\theta) = \mathrm{Tr}(\Sigma_1(\theta)G(\theta))/(2L(\theta)\|G(\theta)\|_F^2) \geqslant \gamma(\theta)$. $\square$

The above proof suggests that the "decoupling" approximation in (2) holds in a weak sense if $\{\|\nabla f(x_i; \theta)\|_{G(\theta)}\}_i$ are uniform. One can apply a similar argument by assuming the uniformity of the fitting errors $\{L_i(\theta)\}$, which, unfortunately, we find never hold in practice. In contrast, we will show that the norm uniformity of model gradients provably holds for linear networks and RFMs, and can be empirically justified for nonlinear networks.

**Over-parameterized linear models.** Consider an over-parameterized linear model (OLM): $f(x; \theta) = F(\theta)^T x$, where $F : \Omega \mapsto \mathbb{R}^d$ denotes a general re-parameterization function. Note that $f(\cdot; \theta)$ only represents linear functions but the corresponding landscape can be highly non-convex. Typical examples include the linear network: $F(\theta) = W_1 W_2 \cdots W_L$ and the diagonal linear network: $F(\theta) = (\alpha_1^2 - \beta_1^2, \ldots, \alpha_d^2 - \beta_d^2)^T$. Both have attracted a lot of attention in analyzing the implicit bias of GD and SGD [3, 42, 33, 13, 4]. The following proposition provides a precise characterization of the noise covariance for OLM models, whose proof is deferred to Appendix D.

**Proposition 2.3.** *Denote by $\mathcal{N}(0, S)$ the Gaussian distribution with mean zero and covariance matrix $S$. Suppose $f(\cdot; \theta)$ is a general OLM and $x \sim \mathcal{N}(0, S)$. Consider the online SGD setting, i.e., $n = \infty$. Then, $\Sigma(\theta) = \nabla L(\theta) \nabla L(\theta)^T + 2L(\theta)G(\theta)$ and $\mu_1(\theta) \geqslant \mu(\theta) \geqslant 1$*

This proposition shows that the alignment property holds in the entire parameter space and moreover, the alignment strength is independent of model size. Here the alignment is only proved for the infinite-sample case. Similar results should hold for finite-sample cases by concentration inequalities as long as $n$ is relatively large, but this straightforward extension does not bring any new insights. It is more interesting to consider the low-sample regime (i.e., $n < d$), where the alignments indeed hold (at least in typical regions explored by SGD) as demonstrated empirically in Figure 1. In addition, the above proposition provides a closed-form expression of the noise covariance, which might be useful for analyzing other properties of SGD beyond the linear stability. A comprehensive analysis of these issues is left to future work.

**Feature-based models.** Consider a feature-based model $f(x; \theta) = \sum_{j=1}^{m} \theta_j \varphi_j(x) = \langle \theta, \Phi(x) \rangle$. In this case, the model gradients: $g_i = g_i(\theta) = \Phi(x_i)$ and the Hessian and Fisher matrix: $G = H = \frac{1}{n} \sum_{i=1}^{n} g_i g_i^T$ all are constant. But the noise covariance $\Sigma(\theta)$ is still state-dependent. The norm uniformity of model gradients also becomes constant: $\gamma = \min_i \chi_i / (\frac{1}{n} \sum_{i=1}^{n} \chi_i)$ with $\chi_i = \|g_i\|_G^2$.

**Lemma 2.4.** $\mu_1(\theta) \geqslant \gamma, \mu_2(\theta) \leqslant \tau(G) := \lambda_1^2(G) / \sum_j \lambda_j^2(G)$, and $\mu(\theta) \geqslant \gamma - \tau(G)$.

The above lemma suggests that the $\mu_2(\theta)$ term is negligible as long as the Fisher matrix is not nearly rank-1. By bounding the $\gamma$ and $\tau(G)$, we can prove the $\mu_1$- and $\mu$-alignment for random ReLU feature models as stated in the following proposition. The proof is presented in Appendix F, where similar results for general RFMs are provided (see Proposition F.7).

**Proposition 2.5.** *If $\varphi_j(x) = \mathrm{ReLU}(w_j^T x)$ with $w_j \overset{iid}{\sim} \mathrm{Unif}(\sqrt{d}\mathbb{S}^{d-1})$ and $x \sim \mathrm{Unif}(\mathbb{S}^{d-1})$. Then, for any $\delta \in (0, 1)$, if $m \geqslant n \gtrsim d^5 \log(1/\delta)$, then w.p. at least $1 - \delta$, $\mu_1(\theta) \geqslant 1$ and $\mu(\theta) \gtrsim d^{-1}$.*

Although feature-based models are linear, their analysis is still applicable to understand nonlinear models, as long as the nonlinear model *locally* behaves like the linearized one: $f_{\mathrm{lin}}(x; \theta) := f(x; \theta^*) + \langle \theta - \theta^*, \nabla f(x; \theta^*) \rangle$ with $\nabla f(x; \theta^*)$ learned from data. Hence, Proposition 2.5 can explain why the alignment property holds in a local region around global minima. Note that this is sufficient for characterizing the linear stability of SGD, which is a local property in nature.

## 2.2 Empirical validations

Figure 1a reports the values of $\alpha(\theta_t), \beta(\theta_t), \mu(\theta_t)$ during the SGD training of four types of models, including the linear networks and RFMs analyzed above, and fully-connected networks (FCN) and convolutional neural networks (CNN). First, one can see that $\alpha(\theta_t)$'s are quite close to 1 during the entire training, suggesting the strong concentration of SGD noise in sharp directions of local landscape. Second, $\beta(\theta_t)$'s keep bounded below, implying the noise magnitudes are sufficiently large with respect to the training loss. As a result, $\mu(\theta_t)$'s are significantly positive for all models examined. In particular, for the linear networks, the alignment holds in the low-sample regime, where $n < d$.

**The size independence.** Figure 1b further examines how the extent of over-parameterization affects the alignment strength. One can see clearly that for linear networks and RFMs, $\mu(\theta)$'s are independent of the model size, which confirms our theoretical findings proved above. In addition, we also observe that for nonlinear networks, the alignment strength is also (nearly) independent of the model size. For instance, for CNN, the value of $\mu(\theta)$ only decreases from around 1.05 to 1.0 as the model size is increased by more than two orders of magnitude.

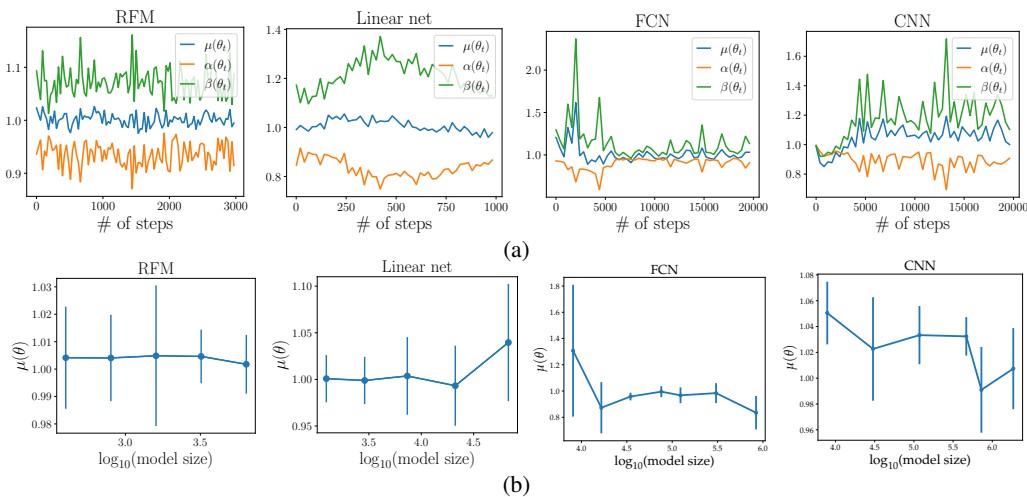

(a)

(b)

Figure 1: **The alignment property of SGD noise**. Four types of models, including the RFM, linear network, fully-connected network (FCN), and convolutional neural network (CNN), are examined. We refer to Appendix A for the experimental setup. Note that the linear network is trained in a low-sample regime ($n = 100, d = 50$). (a) The alignment factors during training. (b) How the alignment strength of convergent solution changes with the over-parameterization. The error bar corresponds to the standard deviation over 5 runs.

**The norm uniformity of model gradients.** Figure 2 shows the norm uniformity of model gradients, where we report the values of $\gamma(\theta)$ in the entire SGD trajectory. On can see that $\gamma(\theta)$ is indeed bounded below, implying that the SGD noise satisfies the $\mu_1$-alignment according to Lemma 2.2.

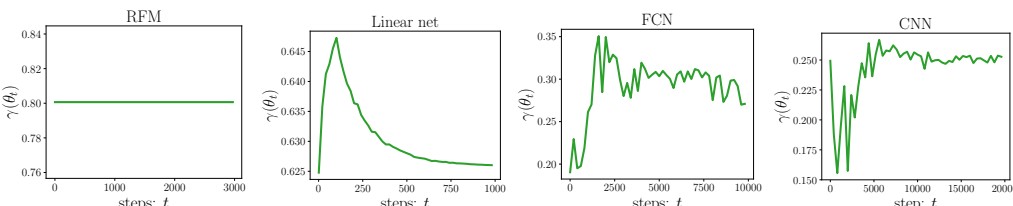

Figure 2: **The norm uniformity of model gradients.** The values of $\gamma(\theta_t)$ in the entire SGD trajectory are reported. It is shown that the norms of model gradient norms are uniform during the entire training process.

Note that in experiments, we only show that the alignment property is satisfied in typical regions explored by SGD, including the random initialization and the convergent region. In contrast, for OLMs and RFMs, we in fact prove the alignment property for the entire parameter space.

## 3 The linear stability analysis

Let $\theta^*$ be a global minimum of $L(\cdot)$. When $\theta_t$ is close to $\theta^*$, the local dynamical behavior of SGD can be characterized by linearizing the dynamics around $\theta^*$:

$$\tilde{\theta}_{t+1} = \tilde{\theta}_t - \frac{\eta}{B} \sum_{i \in I_t} \nabla^2 L_i(\theta^*)(\tilde{\theta}_t - \theta^*), \tag{4}$$

where $\nabla^2 L_i(\theta^*) = \nabla f(x_i; \theta^*) \nabla f(x_i; \theta^*)^T$. This corresponds to the local quadratic approximation of the loss $L(\cdot)$ or the local linearization of the model around $\theta^*$:

$$f_{\text{lin}}(x; \theta) = f(x; \theta^*) + \langle \theta - \theta^*, \nabla f(x; \theta^*) \rangle. \tag{5}$$

Specifically, (4) is exactly the SGD of training the linearized model (5).

**Definition 3.1** (Linear stability). *A global minimum $\theta^*$ is said to be linearly stable if there exists a $C > 0$ such that it holds for the linearized dynamics (4) that $\mathbb{E}[L(\tilde{\theta}_t)] \leqslant C \mathbb{E}[L(\tilde{\theta}_0)], \forall t \geqslant 0$.*

It is well-known in dynamical system that the local behavior of the original nonlinear dynamics can be characterized by the linearized one if the local quadratic approximation is non-degenerate. However, in over-parameterized case, the local quadratic approximation is degenerate in flat directions. Consequently, one may be concerned about the relevance of local quadratic approximation and the

resulting linear stability analysis. Fortunately, the stability in Definition 3.1 is particularly measured with the change of loss value. Thus the instability mostly comes from noise perturbations in sharp directions and the alignment property guarantees that noise mostly concentrates in sharp directions. Consequently, the flat directions contribute little to the instability. In sharp directions, the local quadratic approximation is always valid, thereby explaining the relevance of linear stability analysis. The rigorous formulation of this intuition is left to future work and we instead resort to numerical experiments to demonstrate the validity in this paper.

For simplicity, we will use $\theta_t$ to denote $\tilde{\theta}_t$; let $\theta^* = 0$ and $g_i = \nabla f(x_i; \theta^*)$. For the linearized model $f_{\text{lin}}(\cdot; \theta)$, we have $L(\theta) = \frac{1}{2n} \sum_{i=1}^n |\theta^T g_i|^2 = \frac{1}{2}\theta^T H\theta, G = H = \frac{1}{n}\sum_{i=1}^n g_i g_i^T$, where we omit the dependence on $\theta^*$ for simplicity. Note that the Fisher and Hessian matrix are constant but the noise covariance $\Sigma(\theta) = \frac{1}{n}\sum_{i=1}^n |g_i^T \theta|^2 g_i g_i^T - H\theta\theta^T H$ is still state-dependent.

Before considering the specific linearized SGD (4), we first have a general result.

**Lemma 3.2.** *Consider a general SGD: $\theta_{t+1} = \theta_t - \eta(\nabla L(\theta_t) + \xi_t)$ for the linearized model (5), where $(\xi_t)_{t \geqslant 1}$ are any noises satisfying $\mathbb{E}[\xi_t] = 0, \mathbb{E}[\xi_t \xi_t^T] = S(\theta_t)$. Then we have*

$$\mathbb{E}[L(\theta_{t+1})] = \mathbb{E}[r(\theta_t)L(\theta_t) + \eta^2 v(\theta_t)], \tag{6}$$

*where $\nu(\theta) = \text{Tr}(HS(\theta))/2$ and $r(\theta) \geqslant 0$. Moreover, if $\eta \leqslant 2/\lambda_1(H)$, then $r(\theta) \leqslant 1$.*

*Proof.* Using the fact $\mathbb{E}[\xi_t] = 0$ and $\mathbb{E}[\xi_t \xi_t^T] = S(\theta_t)$, we have

$$\mathbb{E}[L(\theta_{t+1})] = \mathbb{E}[\frac{1}{2}(\theta_t - \eta\nabla L(\theta_t) + \eta\xi_t)^T H(\theta_t - \eta\nabla L(\theta_t) + \eta\xi_t)]$$

$$= \mathbb{E}[L(\theta_t) - \eta\nabla L(\theta)^T H\theta_t + \frac{\eta^2}{2}\nabla L(\theta_t)^T H\nabla L(\theta_t)] + \frac{\eta^2}{2}\mathbb{E}[\text{Tr}(HS(\theta_t))] \tag{7}$$

$$= \mathbb{E}[r(\theta_t)L(\theta_t) + \eta^2\nu(\theta_t)],$$

where $r(\theta) = 1 - 2\eta\frac{\theta^T H^2\theta}{\theta^T H\theta} + \eta^2\frac{\theta^T H^3\theta}{\theta^T H\theta}$ since $\nabla L(\theta) = H\theta$. By Lemma G.2, $r(\theta) \geqslant 0$ and if $\eta \leqslant 2/\lambda_1(H)$, then $r(\theta) \leqslant 1$. ∎

The two terms $r(\theta_t)L(\theta_t)$ and $\eta^2 v(\theta_t)$ denote the contributions from the full-batch gradient $\nabla L(\theta_t)$ and the noise $\xi_t$, respectively. The stability is affected by both terms simultaneously. It is well-known that if $\theta^*$ is linearly stable for GD, then $\lambda_1(H(\theta^*)) \leqslant 2/\eta$ (see, e.g., [46, 31]). This also holds for SGD but SGD imposes a stricter condition because of the extra $\eta^2\nu(\theta_t)$ term. Specifically,

$$\mathbb{E}[L(\theta_{t+1})] = \mathbb{E}[r(\theta_t)L(\theta_t) + \eta^2\nu(\theta_t)] \geqslant \eta^2 \mathbb{E}[\nu(\theta_t)] = 0.5\eta^2 \text{Tr}(HS(\theta)). \tag{8}$$

Therefore, the more $S(\theta)$ aligns with $H$, the more unstable that minimum is. Specifically, let $S(\theta) = 2\sigma^2 L(\theta)C(\theta)$. Then, $\mathbb{E}[L(\theta_{t+1})] \geqslant \eta^2\sigma^2 \mathbb{E}[L(\theta_t) \text{Tr}(HC(\theta_t))]$. Thus to ensure a stable convergence, we should roughly have $\text{Tr}(HC(\theta)) \leqslant \frac{1}{\sigma^2\eta^2}$. We next show that this can lead to a flatness control by utilizing the alignment between $C(\theta)$ and $H$.

## 3.1 The linear stability imposes size-independent flatness constraints

For the mini-batch SGD, the following theorem characterizes how the batch size and learning rate affect the flatness—as measured by the Frobenius norm of Hessian—of minima accessible to SGD.

**Theorem 3.3.** *Let $\theta^*$ be a global minimum that is linearly stable. Denote by $\mu(\theta)$ the alignment factors for the linearized SGD (4) and model (5). If $\mu(\theta) \geqslant \mu_0$, then $\|H(\theta^*)\|_F \leqslant \frac{1}{\eta}\sqrt{\frac{B}{\mu_0}}$.*

*Proof.* By (8) and the definition of $\mu(\theta)$, we have

$$\mathbb{E}[L(\theta_{t+1})] \geqslant \frac{\eta^2}{2B}\mathbb{E}[\text{Tr}(H\Sigma(\theta_t))] \geqslant \frac{\eta^2\|H\|_F^2}{B}\mathbb{E}[\mu(\theta_t)L(\theta_t)] \geqslant \frac{\mu_0\eta^2\|H\|_F^2}{B}\mathbb{E}[L(\theta_t)]. \tag{9}$$

To ensure the stability, we must have $\mu_0\eta^2\|H\|_F^2/B \leqslant 1$, leading to $\|H\|_F \leqslant \sqrt{B/\mu_0}/\eta$. ∎

We have shown in Section 2 that $\mu_0$ is (nearly) size-independent, and thus the obtained upper bound of flatness is also (nearly) size-independent. As a comparison, for GD, the linear stability can only ensure $\lambda_{\max}(H(\theta^*)) \leqslant 2/\eta$. This gives a bound of the Hessian's Frobenius norm: $\|H(\theta^*)\|_F \leqslant 2\sqrt{p}/\eta$,

depending on the model size explicitly. The comparison of two bounds partially explains why SGD tends to select flatter minima than GD.

We show below that the $\mu$-alignment can be further relaxed to the $\mu_1$-alignment. The proof is similar to the one of Theorem 3.3 and deferred to Appendix G.

**Proposition 3.4.** *Under the setting of Theorem 3.3, if the noise of linearized SGD satisfies $\mu_1(\theta) \geqslant \mu_1$, then $\|H(\theta^*)\|_F \leqslant \min\left(\frac{B}{\sqrt{(B-1)\mu_1}}, \frac{2B}{\mu_1}\right)\frac{1}{\eta}$.*

When $B \gg 1$, the bound becomes $B/(\eta\sqrt{(B-1)\mu_1}) \approx \sqrt{B/\mu_1}/\eta$, which is the same as the case of $\mu$-alignment. Thus the influence of $\nabla L$ is indeed negligible compared with $\{\nabla L_i\}_i$.

Note that the linear stability is local in nature and hence our analysis essentially only needs the $\mu_1$-alignment to hold locally around minima of interest. Experiments in Figure 2 shows that $\gamma(\theta^*)$ is always bounded below, i.e., the norm uniformity of model gradients is satisfied. Combining with Proposition F.7, we can conclude that the alignment property assumed in Proposition 3.4 holds.

**Tightness of our analysis.** In the analysis above, we only inspect the instability caused by the noise, with the full-batch gradient completely ignored. Therefore, we anticipate that our bound is tighter in small-batch regime, where the noise term dominates the full-batch term. We will see that our numerical experiments indeed confirm this tightness in small-batch regime. However, for obtaining the tightest bound, one may need to consider both components simultaneously; this is much more complicated and left to future work.

**Numerical validations.** Figure 3a numerically shows how the Frobenius norm of Hessian (not only the upper bound) changes with extent of over-parameterization, where the trace of Hessian is also plotted for comparison. One can see that the Frobenius norm indeed keeps almost unchanged as increasing the model size but the Hessian trace increases significantly. Figure 3b further shows the ratio between the Frobenius norm and our upper bound in the training process and two batch sizes are examined. We have two observations. First, the correctness of our bound holds for the entire SGD trajectory, suggesting that the linear stability analysis is relevant for the entire training process. Second, as expected, the theoretical bound is indeed tighter for the case with a smaller batch size.

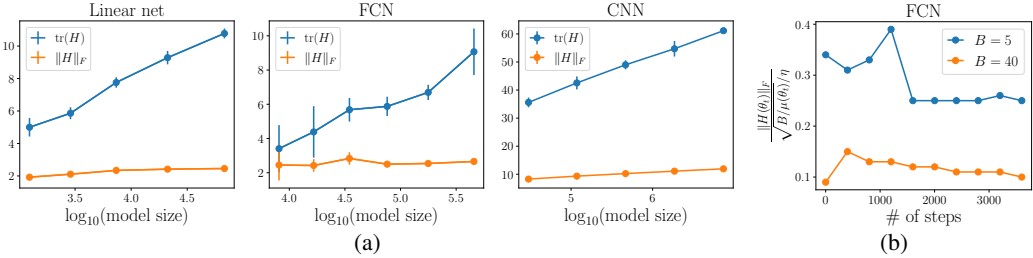

Figure 3: (a) The Frobenius norm and trace of Hessian vs. model size. The error bar corresponds to the standard deviation estimated over 5 runs. (b) The ratio between the Frobenius norm and our upper bound in the training process. Here $B = 5$ and $B = 40$ are examined. For more experiment details, we refer to Appendix A.

## 3.2 SGD escapes from sharp minima exponentially fast

The following theorem shows that the pure noise-driven escape from a sharp minimum is *exponentially fast*, whose proof follows trivially from (9).

**Theorem 3.5.** *Under the setting of Theorem 3.3, if $\|H(\theta^*)\|_F > \frac{1}{\eta}\sqrt{\frac{B}{\mu_0}}$, then the linearized SGD satisfies $\mathbb{E}[L(\theta_t)] \geqslant \gamma_0^t \, \mathbb{E}[L(\theta_0)]$ with $\gamma_0 = \frac{\eta^2\mu_0}{B}\|H(\theta^*)\|_F^2 > 1$.*

Hence, linearized SGD takes roughly $\log_{\gamma_0}(1/\varepsilon)$ steps to escape from a $O(\varepsilon)$-loss region to a $O(1)$-loss region. The escape time depends on the loss barrier only logarithmically and is independent of the parameter space dimension. Due to the local closeness between linearized SGD and the original SGD, this partially explains the unreasonable escape efficiency of SGD for training big models. In contrast, the escape rates of existing works [48, 52, 51, 30] are either exponentially slow with respect to the loss barrier or suffer from the curse of dimensionality.

Figure 4 shows the trajectories of SGD escaping from sharp minima. It is demonstrated that the escape is indeed exponentially fast and specifically, 10 steps are enough for SGD escaping to a high-loss region for all the models examined. In addition, we observe that the escape is still exponentially fast in the high-loss region, although our analysis only applies to a local region. How can we explain this nonlocal escape behavior? We leave the study of this interesting phenomenon to future work.

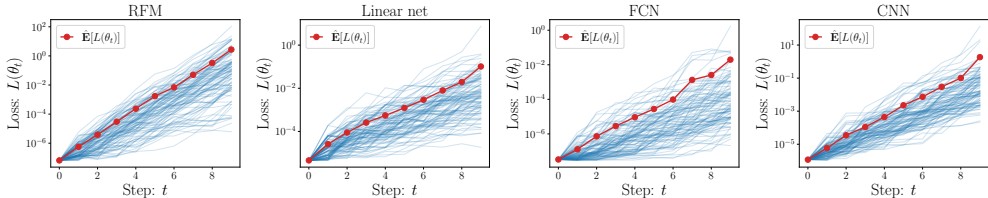

Figure 4: **The exponentially fast escape from sharp minima.** The blue curves are 200 trajectories of SGD; the red curve corresponds to the average. The sharp minimum is found by GD. When GD nearly converge, we switch to SGD with the same learning rate. This choice ensures that the minimum is stable for GD, and thus the escape is purely driven by SGD noise. For more experimental details, we refer to Appendix A.

### 3.3 The importance of the noise structure

**The magnitude structure.**   Theorem 3.5 together with its proof suggests that the loss dependence of noise magnitude is critical for obtaining the exponentially fast escape. The intuition is as follows. When $\theta_t$ is perturbed by noise to $\theta_{t+1}$ where $L(\theta_{t+1}) > L(\theta_t)$, the noise magnitude becomes larger there and thus $\theta_{t+1}$ is easier to be perturbed to a larger-loss region. This positive feedback drives SGD to leave exponentially fast. On the contrary, the following lemma shows that if the noise is uniformly bounded, the noise-driven escape is at most linear in time.

**Lemma 3.6.** *Under the setting of Lemma 3.2, assume $\eta \leqslant 2/\lambda_1(H)$ and $\mathbb{E}[HS(\theta)] \leqslant 2\sigma^2$. Then $\mathbb{E}[L(\theta_t) - L(\theta_0)] \leqslant \eta^2\sigma^2 t$.*

We set $\eta \leqslant 2/\lambda_1(H)$ to avoid the exponential escape caused by the full-batch component.

*Proof.* By Lemma G.2, when $\eta \leqslant 2/\lambda_1(H)$, $r(\theta) \leqslant 1$. Thus Lemma 3.2 implies $\mathbb{E}[L(\theta_{t+1})] \leqslant \mathbb{E}[L(\theta_t)] + \eta^2\sigma^2$, which implies $\mathbb{E}[L(\theta_t)] \leqslant \mathbb{E}[L(\theta_0)] + \eta^2\sigma^2 t$. $\square$

**The direction structure.**   We now turn to consider the impact of direction structure. Consider general SGDs: $\theta_{t+1} = \theta_t - \eta(\nabla L(\theta_t) + \xi_t)$ with $\mathbb{E}[\xi_t\xi_t] = S(\theta_t)/B$ for the linearized model (5). We compare two type of (unrealistic) noises:

- *Geometry-aware noise:* $S_1(\theta) = 2L(\theta)H$.
- *Isotropic noise:* $S_2(\theta) = 2\sigma^2 L(\theta)I_p$ with $\sigma^2 = \text{Tr}(H)/p$. Here, the value of $\sigma^2$ is chosen to ensure two types of noises have the same total variance for a fair comparison [52].

Note that $p$ denotes the model size. Analogous to Theorem 3.3, for the second isotropic SGD,

$$\mathbb{E}[L(\theta_{t+1})] \geqslant \frac{\eta^2}{2B}\text{Tr}(HS(\theta)) \geqslant \mathbb{E}[L(\theta_t)]\frac{\sigma^2\eta^2}{B}\text{Tr}(H) = \mathbb{E}[L(\theta_t)]\frac{\eta^2}{pB}\text{Tr}(H)^2.$$

Hence, the instability decreases with the parameter-space dimension and the resulting flatness constraint is $\text{Tr}(H(\theta^*)) \leqslant \sqrt{pB}/\eta$, depending on the model size explicitly. In contrast, for the fist noise, Theorem 3.3 implies $\|H(\theta^*)\|_F \leqslant \sqrt{B}/\eta$, independent of the model size. This difference can be intuitively explained as follows. The isotropic noise wastes most energy in perturbing SGD along flat directions, which barely affects the instability. In contrast, the geometry-aware noise focuses most energy on perturbing SGD along sharp directions, causing much more instability.

## 4   Larger-scale experiments

We have provided small-scale experiments to justify the validity of our theoretical findings for a variety of ML models. Here we turn to demonstrate the practical relevance by consider the classification of the CIFAR-10 dataset [22] with VGG nets [36] and ResNets [15]. In training, all explicit regularizations are removed to keep consistent with our theoretical analysis. More details of the experimental setup can be found in Appendix A.

**The alignment property and escaping behavior.** Figure 5a reports the alignment strength of SGD noise during training for VGG-19 and ResNet-110. One can see that the alignment factors are significantly positive and similar results are also observed for a variety of VGG nets and ResNets of different depths and can be found in Figure 8 of Appendix B. On can see that the alignment strength is nearly independent of the model size. Figure 5b shows the behavior of escaping from sharp minima for VGG-19 and ResNet-110. One can still observe that the escape is exponentially fast and similar observation for other ResNets and VGG nets can be found in Figure 9 in the appendix. These observations suggest that our theoretical findings also hold for this practical setting.

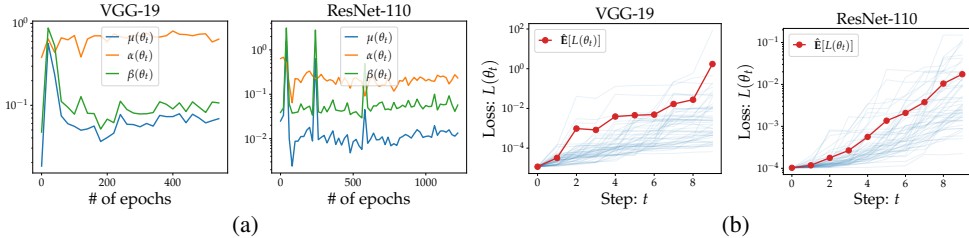

Figure 5: (a) The alignment factors and the escaping behavior for VGG-19. Similar results are also observed for all examined ResNets and VGG nets, which can be found in Appendix B.2. (b) The actual flatness of SGD solution and the corresponding theoretical upper bound (Theorem 3.3). (c) The upper bound becomes tighter as decreasing the batch size.

**The flatness and upper bound.** Figure 6a reports the flatness of convergent solution and the corresponding upper bound predicted by our theory for ResNets and VGG nets. It is again observed that the flatness is nearly independent of the model size. A surprising observation in Figure 6a is that our upper bounds are rather tight, see, e.g., VGG-16 and VGG-19. This tightness suggests that SGD runs (nearly) at the edge of stability [46, 8]. Moreover, as expected, Figure 6b shows that the our bound becomes tighter as decreasing batch size. This is consistent with what we observe in small-scale experiment in Figure 3b.

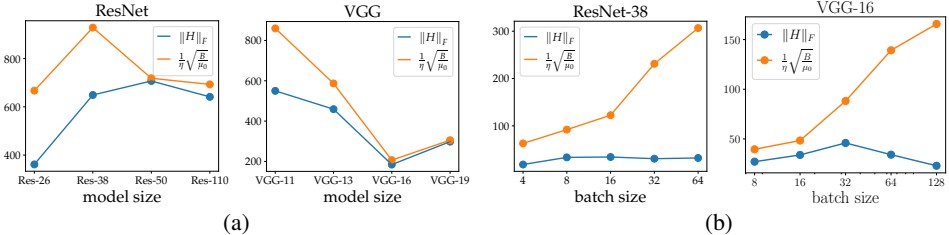

Figure 6: (a) The flatness and upper bound for various ResNets and VGG nets. (b) How the tightness of upper bound changes with decreasing batch size.

## 5 Conclusion

We provide a stability-based explanation of why SGD selects flat minima. Our current understanding is as follows. 1) For popular ML models, the SGD noise aligns very well with local landscape. 2) This alignment property ensures that the flatness of stable minima must be size-independent. This understanding is made rigorous and quantitative by introducing a loss-scaled alignment factor to characterize the alignment strength and analyze the linear stability. Obviously, many questions remains open. For example, can we understand what roles the stability plays in the whole dynamic process of SGD instead of only around global minima? Can we establish the connection between the Hessian's Frobenius norm and generalization? Can we provide a fine-grained characterization of the noise structure and how the structure is related to implicit regularization of SGD? We leave the discussion of these important questions to future work.

### Acknowledgements

We thank Zhiqin Xu and the anonymous reviewers for helpful suggestions. The work of Lei Wu is supported by a startup fund from Peking University. The work of Weijie J. Su is supported in part by NSF Grants CAREER DMS-1847415 and an Alfred Sloan Research Fellowship.

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
