# OpenReview forum: "The alignment property of SGD noise and how it helps select flat minima: A stability analysis"
_NeurIPS.cc/2022/Conference — NeurIPS 2022 Accept_

### Official Review · Reviewer_9D5P · 2022-07-07

**Rating:** 7
**Confidence:** 4
**Soundness:** 4 excellent
**Presentation:** 3 good
**Contribution:** 3 good

**Summary:**

Deep learning folklore holds that the gradient noise in SGD causes it to prefer "flat minima."  An important open question in deep learning theory is to make this folklore mathematically precise.  Taking a step in that direction, this paper describes a mechanism by which SGD escapes exponentially quickly from minima where the Frobenius norm of the Hessian is too large.  This mechanism is orthogonal to the curvature-driven process that causes full-batch gradient descent to escape from minima where the Hessian spectral norm exceeds 2 / step size.

**Questions:**

-- I didn't understand why the submission keeps mentioning that the flatness is independent of model size.  Could the authors clarify why this is an important point?

-- Are the authors sure that the linear stability analysis is only valid near a local minimum?  I ask because in the full-batch case, the curvature-driven escape when sharpness exceeds 2 / (step size) is valid generically, not just near a local minimum.

-- I wonder if the bound would be tighter on the real networks (VGG, ResNet) if you considered a batch size smaller than 64.  Intuitively, I would expect the noise-driven escape to dominate when the batch size is small.

**Limitations:**

Yes, the authors adequately addressed the limitations.

**Strengths And Weaknesses:**

Strengths:

  -- To the best of my knowledge, the idea of an exponentially fast escape that is driven purely by noise is novel.  It is interesting that a required condition for this phenomenon (noise magnitude proportional to loss value) is provably satisfied in linear nets and random feature models.

Weaknesses:

  --  The analysis only yields a sufficient condition for instability, not a necessary-and-sufficient condition.

  -- While in general the stability will depend on both the full-batch component and the noise component, the analysis here considers only the noise component in isolation.  Accounting for both simultaneously is going to be hard, so I don't begrudge the authors for this simplification.

  -- I think the paper would be clearer if the authors first presented the escape analysis (section 3) before the sufficient conditions (section 2).  When I read section 2, I spent a lot of time scratching my head wondering why alpha, beta, and mu are defined the way they are.  Later, when I got to section 3, I realized that \mu is precisely what is needed to trigger exponentially fast escape.

---

> ### Author Response · Authors · 2022-08-02
> **Why the submission keeps mentioning that the flatness is independent of model size.**
>
> A short explanation: The size-independence of flatness is crucial for ensuring the complexity of SGD solutions to be bounded independently of the model size, thereby ensuring the generalization in the over-parameterized regime. A detailed explanation is given below.
> 1. First, if we view the flatness, which is Hessian's fro-norm in the current paper, as a complexity measure that can effectively control the model's capacity, then the size independence implies that the complexity of SGD solutions does not increase as increasing the model size. Therefore, it is crucial for arguing that SGD finds generalizable solutions, in particular in the over-parameterized regime. Specifically, in a separate work (we will appropriately cite it  in the final version of this submission), we already proved that for two-layer neural nets (under certain conditions),
>
> $$
>   \qquad\qquad\qquad \qquad gen-gap (\theta^*) \leq O\left(\frac{||H(\theta^*)||_F}{\sqrt{n}}\right) 	\qquad \qquad \qquad \qquad \qquad (P),
> $$
>
> where the hidden constant only depends on the input dimension linearly. Together with the size independence of $||H(\theta^*)||_F$ guaranteed by the linear stability,  we can conclude that SGD finds generalizable solutions, thereby explaining the **implicit Regularization**.
>
> 2. Secondly, this size independence of Hessian's Fro-norm is also a major difference between SGD and GD. For GD, the stability only guarantees that $\lambda_\max(H(\theta^*))\leq 2/\eta$. Therefore, in general, GD finds minima, where only the curvature of the sharpest direction is controlled. By contrast, SGD tends to find minima, where the landscape is uniformly flat in different directions due to that the whole spectrum of Hessian is controlled. A naive bound of the Hessian's Fro-norm of GD solution is
>
> $$
> \qquad\qquad\qquad ||H(\theta^*)||_F\leq \sqrt{\text{rank}(H(\theta^*))}  \lambda_\max (H(\theta^*))\leq \frac{2\sqrt{\min(n,p)}}{\eta},
>   $$
>
> where $n,p$ denotes the sample and model size, respectively. Plugging this into the preceding generalization bound (P) only yields a vacuous/trivial bound of generalization gap. This partially explains why SGD generalizes better than GD.
>
> In a word, the size independence of Hessian's Fro-norm is critical to explaining why SGD selects generalizable minima and distinguishes SGD from GD.

---

> > ### Author Response · Authors · 2022-08-02
> > **I wonder if the bound would be tighter on the real networks (VGG, ResNet) if you considered a batch size smaller than 64.**
> >
> > Yes, you are absolutely correct. We have done some extra experiments  to see how the batch size affects the tightness of our bound and the experiment results (given by the two table below) confirm your conjecture. We also refer the reviewer to the descriptions in lines 329-336 and Figure 5c of the revised submission for more details.  Note that due to the page limit, the result of VGG16 is not added into the revisioned submission. Notice that here we only conduct the experiment for classifying a two-class subset of the CIFAR-10  due to the time limit. We will update it to the full CIFAR-10 experiment in the future.
> >
> >
> > &nbsp;
> >
> > **ResNet 38**
> >
> > | __batch size__ | 4 | 8 | 16 | 32 | 64 |
> > |-|-|-|-|-|-|
> > | __bound__ | 62.92 | 92.02 | 122.48 | 231.13 | 306.64 |
> > | __flatness__ | 18.34 | 33.38 | 34.18 | 30.61 | 32.15 |
> >
> > &nbsp;
> >
> > **VGG16**
> >
> > | __batch size__ | 8 | 16 | 32 | 64 | 128 |
> > |-|-|-|-|-|-|
> > | __bound__ | 39.59 | 48.41 | 88.18 | 139.16 | 165.71 |
> > | __flatness__ | 27.11 | 33.90 | 45.95 | 34.18 | 23.01 |

---

> ### Author Response · Authors · 2022-08-02
> **Are the authors sure that the linear stability analysis is only valid near a local minimum?**
>
> Indeed, in this paper, we only performed the linear stability analysis (LSA) near global minima, but we do not suggest that LSA is irrelevant in other regions, in particular in the SGD training. Define the following stability factor at the SGD solution $\theta_t$ by
> $$
> \qquad \qquad \qquad \qquad \gamma_t =\frac{ \sqrt{B/\mu(\theta_t)}/\eta}{||H(\theta_t)||_F}.
> $$
> The condition of linear stability   is $\gamma_t\leq 1$. The following table reports the values of $\gamma_t$'s in SGD trajectory for the case of batch size=5 (second column) and batch size=40 (third column).  Here the model is the fully connected networks with other settings same as the ones provided in the submission.
>
> | steps | B=5  | B=40 |
> | ----- | ---- | ---- |
> |1|0.34| 0.09|
> |401|0.31| 0.15|
> |801|0.33| 0.13|
> |1201|0.39| 0.13|
> |1601|0.25| 0.12|
> |2001|0.25| 0.12|
> |2401|0.25| 0.11|
> |2801|0.25| 0.11|
> |3201|0.26| 0.11|
> |3601|0.25| 0.10|
>
> We see clearly that the $\gamma_t<1$, i.e.,  Hessian's Fro-norms are smaller than the bounds predicted by LSA, during the whole SGD training. Moreover, as expected, the bounds of the small-batch case are tighter. We thus can conclude that SGD stays in the stable region predicted by LSA.
>
> Theoretically speaking, the validity of LSA near fixed points (i.e., global minima) only requires that the landscape **locally** behaves like a quadratic function, which is often true since the first-order coefficients are zeros at global minima. However, *for non-fixed points, the validity of LSA needs the landscape to behave like a quadratic function in a **non-local** scale*, at least in the most-unstable directions (typically, the leading eigen directions of Hessian).
>
> However, due to the page limit, we leave the discussion of the implications and explain why LSA is valid beyond global minima, i.e., justifications of the non-local quadratic behavior of neural network landscape, to future work.

---

> ### Author Response · Authors · 2022-08-02
> **Other questions**
>
> **Q: The analysis only yields a sufficient condition for instability, not a necessary-and-sufficient condition.**
>
> Yes, you are correct. To obtain a sufficient-and-necessary condition, we must consider both the curvature-induced and noise-induced instability simultaneously. However, this is rather complicated, e.g., we may need to track the instabilities along different eigen directions instead of only the average ones as done currently. For this reason, we decided to leave this to future work.
>
> On the other hand, we stress that our condition is **necessary for stability**, which allows us to conclude that SGD tends to select flat minima (under the condition that the noise aligns with the local landscape). This is one of the key differences/improvements over (Wu et al., 2018).
>
> &nbsp;
> &nbsp;
> ---
>
> **Q: While in general the stability will depend on both the full-batch component and the noise component, the analysis here considers only the noise component in isolation.**
>
> We have added a remark on this point in lines 265-268 in the revised submission.

---

### Official Review · Reviewer_jEC4 · 2022-07-11

**Rating:** 6
**Confidence:** 4
**Soundness:** 3 good
**Presentation:** 3 good
**Contribution:** 3 good

**Summary:**

The authors of the paper present the following contributions:
- They provide an in-depth study of the importance of the SGD noise both in term of geometry and in terms of scale.
- They show that, if a certain alignment property is satisfied, a global minimum is linearly stable if and only if the Froebinius norm of the Hessian at optimum is upper bounded by constant independent of model and sample size

**Questions:**

In my opinion, the crux of the paper is to give a precise (or experimental) sense to the approximations of equation (2). Maybe the authors could try to provide estimation of how far each approximation is to the real covariance:
-  First regarding the influence of the loss
- Second regarding the independence between $\nabla f(x_i, \theta)$ and $L_i$

This corresponds to constants $\mu_1$ and $\mu_2$ defined in the paper but no experiments are shown concerning these variables.

**Limitations:**

I use this paragraph to conclude as I already discussed the limitations on the previous boxes. The paper tries to conduct both theoretical and experimental explanations of the wide minima selection of SGD. I'll weakly accept the paper for its clarity and detail about the noise. To raise my score, I would like the authors to justify more the fact that the escaping phenomenon is important and the alignment phenomenon of the covariance structures.


**Strengths And Weaknesses:**

### **Strengths**

- The first thing I have to say is that the paper is very well written. The exposition is clearly conducted and all assumptions and estimations are detailed, announced and commented. One may or may not validate the authors' noise model, but at least it is not hidden (as too often) and keys are given to appreciate the results
- The main strength and difference of the article is the particular attention given on the noise model both in terms of noise and of geometry. From this, even if the results seem easy to prove, the exposition is cristal clear and more convincing with the previous literature.

### **Weaknesses**

- There are a lot of results and approximation stated in the article, but overall, the crux of the paper is two show that the approximations of the equation (2) at the beginning of the paper is valid: that is to say that either $\alpha, \beta$ or $\mu$ are close to $1$. If Figure 1 is  pretty convincing from this point of view, I have to say that Figure 5 is not as the scale of these constant can be (way) smaller than 1. I would really appreciate if the authors comment more on this point because the sentence "*This comparison suggests that the alignment strength significantly depends on the intrinsic complexity of the problem, (nearly) independent of the model size*" is not very convincing.
- This is a minor weakness but overall, even if the paper is not overselling and clear about their study, I am still not convinced by the stability argument of SGD. Indeed, on never see plots like Figure 4 in real training dynamics of neural networks and this suggests -at least to me- that the taking into account the noise is not a stability issue but a dynamical one. Maybe the authors could comment a bit on this fact.


### **Minor typos**

- lign 65: $\lambda_i$ and not $\lambda_1$
- lign 227: In **the** current paper
- ligns after 235: there is a confusion between $v$ and $\nu$

---

> ### Author Response · Authors · 2022-08-02
> **"This comparison suggests that the alignment strength significantly depends on the intrinsic complexity of the problem, (nearly) independent of the model size" is not very convincing.**
>
> We completely agree that this claim is not well supported and even not accurate. We have rewritten this sentence and added more experiments and discussions; we refer the reviewer to lines 337-346 (marked in blue) in the revised submission. The following tables show how the values of $\alpha,\beta, \mu$ of convergent solutions change when increasing of the number of classes. Here, we only report the alignments at convergence for simplicity, and we checked that similar patterns also hold during the training. One can see a  clear trend that $\beta$ and $\mu$ decrease with the number of classes, but the angle alignment $\alpha$  is more robust with the number of classes. These results have not been added to the revised paper only because of the page limit.
>
> VGG16 for classifying MNIST
>
> | __#class__ | 2 | 5 | 8 | 10 |
> |-|-|-|-|-|
> | $\mu$ | 0.22 | 0.17 | 0.11 | 0.10 |
> | $\alpha$ | 0.79 | 0.71 | 0.86 | 0.82 |
> | $\beta$ | 0.28 | 0.24 | 0.12 | 0.12 |
>
>
> VGG16 for classifying CIFAR-10
>
> | __#class__ | 2    | 5    | 8    | 10   |
> | ---------- | ---- | ---- | ---- | ---- |
> | $\mu$      | 0.85 | 0.14 | 0.08 | 0.06 |
> | $\alpha$   | 0.91 | 0.87 | 0.73 | 0.81 |
> | $\beta$    | 0.94 | 0.17 | 0.10 | 0.08 |
>
> However, obtaining some conclusive results still needs much more work, such as discussing the influence of model architectures. Considering the page limit of NeurIPS, we prefer to leave the systematical study to future work.

---

> ### Author Response · Authors · 2022-08-02
> **the crux of the paper is to show that the approximations of the equation (2) at the beginning of the paper is valid:**
>
> The decoupling approximation is introduced to heuristically explain the geometry awareness of SGD noise and to motivate our definitions of alignment factors. We stress that we are not trying to say that the decoupling approximation itself is valid.
>
> 1. *In our opinion, instead of showing the validity of the decoupling approximation in equation (2), we relax it to the* nondegeneracy *of the alignment between the noise covariance and local Hessian.* The latter is much weaker than the former, which implies a complete alignment. Note that the alignment nondegeneracy has been numerically verified for practical models and theoretically justified for some toy models. In contrast, the former, i.e., the decoupling approximation, is always invalid. For example, the cifar10 experiments **numerically** show that $\alpha(\theta), \mu(\theta)$ is not close to $1$; Theorem 2.1 shows that the decoupling approximation **provably** loses a rank-1 term for linear networks.
>
> 2.  However, we do agree with the reviewer that a fine-grain characterization of the noise covariance in a  strong sense is also important. One particular example is mentioned by the reviewer: Why the alignment factors are close to 1 for small-scale problems but kind of away from $1$ for the CIFAR-10 problem? A systematical study of how different factors, such as model architectures, model size, and sample size, affect the alignment strength is beyond scope of this paper. We leave it to future work. Here, we particularly mention that the boundedness of the alignment defined by us only implies an average concentration between the noise covariance and local Hessian. As a consequence, we can only show that the expected loss blows up exponentially in the escape process. If we want to characterize properties such as the escape directions, a stronger characterization of the noise covariance will be needed.
>
> Lastly, we emphasize that the most important contribution of this paper is to show that stability can ensure that SGD only selects flat minima if the SGD noise aligns with the local landscape. A more fine-grained characterization of SGD noise is definitely important and helpful for understanding SGD dynamics but beyond the scope of the current paper.
>
>
>
> &nbsp;
> &nbsp;
> ---
> **Q: Maybe the authors could try to provide estimation of how far each approximation is to the real covariance.**
>
> We feel that this is beyond the scope of this paper, though we agree that this is a very interesting and important question. Note that the analysis of (average) linear stability is valid as long as the alignment is non-degenerate. In other words, our stability analysis works even if the decoupling approximation is invalid. Therefore, estimating the error of decoupling approximation is irrelevant. More importantly,  one of the most important messages hidden in our analysis is: It is unnecessary to pursue a precise/strong characterization of the noise covariance for analyzing some properties of SGD; certain weak characterizations such as the alignments are sufficient.
>
>
> &nbsp;
> &nbsp;
> ---
> **Q: In Figure 1, $\alpha, \beta$ and $\mu$ or  are close to 1. Figure 5 is not as the scale of these constants can be (way) smaller than 1.**
>
> The finding that these alignments, in particular the angle alignment $\alpha$, are close to $1$ for those small-scale experiments is striking and unexpected; it might be very important for understanding SGD dynamics. However, as explained above, in this paper, we only need that $\mu$ to be bounded below instead of close to 1 and this has been sufficiently supported by current experiments.

---

> ### Author Response · Authors · 2022-08-02
> **I am still not convinced by the stability argument of SGD. one never see plots like Figure 4 in real training dynamics of neural networks.**
>
> First, we would like to point out that the escape phenomenon in Figure 4 does occur in real training of neural networks. This is the typical case of training with a *cyclical learning rate (LR)*. We refer the reviewer to Figure 2 in (Smith, 2017), Figure 2 in (Huang et al., 2017), and Figure 2 in (Izmailov et al., 2018), where one can see that the training loss **suddenly** increases (within a few iterations) to $\Omega(1)$ when increasing the learning rate. This intriguing phenomenon in cyclical LR training can be explained by the exponential escape behavior investigated in our paper. We believe that one can at least partially explain the implicit regularization of cyclical LR training by using the stability argument, which is one of our ongoing projects.
>
> Second, even for the normal training of neural networks, the stability argument is still relevant. It can explain why SGD does not enter sharp regions during the whole training process since SGD is (exponentially) unstable there. Specifically, in the current paper, we only focus on the end of training (i.e., around global minima ). We also refer the reviewer to our response to Reviewer 9D5P (click [the link](https://openreview.net/forum?id=rUc8peDIM45&noteId=nIC2VFvJYnY)), where we provide a preliminary experiment showing that the Hessian's Fro-norm is below the upper bounded predicted by the stability argument during the entire SGD trajectory. We also kindly refer the reviewer to  (Cohen et al., 2021), which studies the whole GD trajectory using linear stability and finds that the leading eigenvalue of Hessian is close to 2/learning_rate.  All these indicate that the stability argument is also relevant for studying the training process. However, a systematic study needs much more work, which we leave to future work.
>
> Third, stability arguments can also explain why SGD/GD solutions generalize well for some simplified models. For example, recent work (Mulayoff et al., 2021; Nacson et al., 2022) shows that the largest eigenvalue of Hessian can bound the generalization gap for *univariate* two-layer networks and two-layer diagonal linear networks. Combined with the stability condition of GD, these works imply that GD only selects generalizable minima. For SGD, (Ma et al., 2021) shows that the linear stability can ensure the boundedness of the Sobolev seminorm of the implemented functions for MLP, thereby explaining the generalization. However, it is well-known that the boundedness of the Sobolev seminorm cannot explain the generalization in high dimension, since the corresponding generalization bound suffers from the curse of dimensionality. In a separate work (we will appropriately cite it in the final version of this submission), we already proved that the Fro-norm of Hessian can effectively control the generalization gap of two-layer neural networks (2LNN); see our response to Reviewer *9D5P* on the importance of size independence of flatness (click [the link](https://openreview.net/forum?id=rUc8peDIM45&noteId=82YsEyHgU0F)). Combing with the stability analysis in the current paper, we can conclude that SGD only selects minima that provably generalize for 2LNN.
>
> We emphasize that stability analysis is a simple but very powerful tool to analyze general nonlinear dynamics. The major advantage of stability analysis is its generality. For example, our linear stability analysis applies to the training of real deep networks and yields meaningful characterizations of the dynamical behavior of SGD, e.g., explaining the selection of flat minima. In contrast, the analysis that relies on precise descriptions of the dynamic processes only works for some toy models, such as linear networks.
>
> Lastly, we do agree with the reviewer that many issues cannot be explained by merely the stability argument. For example, the stability argument cannot give us some precise descriptions of the training process, such as the convergence rate.
>
> &nbsp;
> ---
> **Reference**
>
> Leslie N. Smith, Nicholay Topin, *Exploring loss function topology with cyclical learning rates*, ICLR 2017 workshop track.
>
> Gao Huang, et al., *Snapshot Ensembles: Train 1, get M for free*, ICLR 2017
>
> Pavel Izmailov, et al.,  *Averaging Weights Leads to Wider Optima and Better Generalization*, UAI 2018
>
> Jeremy M. Cohen, et al., *Gradient Descent on Neural Networks Typically Occurs at the Edge of Stability*, ICLR 2021
>
> Rotem Mulayoff, et al., *The implicit bias of minima stability: A view from function space*, NeurIPS 2021
>
> Mor Shpigel Nacson,  et al., *Implicit Bias of the Step Size in Linear Diagonal Neural Networks*, ICML 2022
>
> Chao Ma, et al., *On linear stability of SGD and input-smoothness of neural networks*, NeurIPS 2021

---

> > ### Comment · Reviewer_jEC4 · 2022-08-02
> > **Rebuttal**
> >
> > Thanks for pointing out precise training dynamics in the case of cyclical step sizes.
> >
> > For the other arguments on linear stability, I understand the simplicity of such an analysis. However, I am sorry to say that I am still not convince neither by the arguments presented in the comments nor by the cited papers that I find fuzzy on numerous points.
> >
> > For the other points, I thank the authors for the answer and will not change my score.

---

> > > ### Author Response · Authors · 2022-08-02
> > > **Could you be specific on the points that you find fuzzy and not convincing?**
> > >
> > > Dear reviewer,
> > >
> > > Thanks for your quick feedback. We would appreciate it if you could be more specific on the points that you find not convincing.

---

> > > > ### Comment · Reviewer_jEC4 · 2022-08-07
> > > > **Stability analysis further thoughts**
> > > >
> > > > Dear authors,
> > > > this was simply a conclusive thought on linear stability. Obviously this is a powerful tool and the linear approximation allows gently to derive theorems and provide some *necessary* conditions for convergence. However, to detail my thinking, I have to say that these stability analysis seem to provide only a small picture on why GD/SGD generalises well: for example, I know well a lot of the cited papers of the rebuttal and they are all written *as if* the trajectory implicitly minimises the curvature of the loss (stability criterion) to enhance stability. This is absolutely not the case, and at a given step-size there are a lot of stable global minima that generalise poorly. On this perspective the presented paper is less overclaiming than the cited ones.

---

> > > > > ### Author Response · Authors · 2022-08-09
> > > > > **Stability can distinguish flat minima from sharp minima. It is important if one agrees that flat minima argument is important.**
> > > > >
> > > > > Dear Reviewer,
> > > > >
> > > > > We truly appreciate your comment and partially understand your considerations. However, we respectfully disagree with you on most points as explained below.
> > > > >
> > > > > ---
> > > > >
> > > > >  > "stability analysis seems to provide only a small picture on why GD/SGD generalises well"
> > > > >
> > > > >    We agree that the stability analysis provides only a **small picture** of the implicit regularization. However, this does not mean that the stability analysis is *not convincing and relevant*. A detailed characterization of one of the small pictures  can be critical for revealing and  understanding the big picture, let alone that the big picture has not emerged.
> > > > >
> > > > >
> > > > >    We clearly demonstrate that stability plays an critical role in explaining why SGD selects flat minima in the large learning rate (LR) and small batch regime. If one agrees that the **flat minima argument** is important, then the stability analysis presented here is obviously useful. However, the flat minima argument itself is sort of not convincing. From a theoretical perspective, we agree on this point since most existing arguments of the generalization of flat minima are rather hand-waving not sufficiently scientific. Even so, the practical experiments already sufficiently justified that the relevance of flat minima argument. Therefore, in our opinion, the importance and relevance currently are supported empirically instead of theoretically.
> > > > >
> > > > > &nbsp;
> > > > >
> > > > >
> > > > >
> > > > >  > "they are all written as if the trajectory implicitly minimises the curvature of the loss (stability criterion) to enhance stability.""
> > > > >
> > > > > To be honest, in our understanding,  existing related works did not claim that the SGD/GD trajectory implicitly "minimizes" curvatures of the loss landscape. The stability only ensures that SGD stays in a flat region where the stability condition is satisfied. In other words, the stability imposes constraints on loss curvatures instead of actively minimizing them. In fact, as shown in (Cohen et al., ICLR 2021), **GD dynamics itself, in fact, keeps progressively increasing the curvature until the edge of stability is reached**, after which the curvature hovers there and does not increase any more because of the stability constraint. This increasing curvature nature of GD provides an explanation of  why the flatness of GD solutions are close to  2/learning_rate instead of  away from it, after all the stability condition is only necessary. We suspect that similar phenomena also happen to SGD but need much more investigations, which we leave to future work.
> > > > >
> > > > > &nbsp;
> > > > >
> > > > > > "given step-size there are a lot of stable global minima that generalise poorly"
> > > > >
> > > > > In general, this might be true and specifically, we can detail the situations where this happens as follows.
> > > > >
> > > > >   - When the learning rate is infinitesimal, SGD becomes gradient flow; obviously, all minima are stable now but there definitely exist ones that generalize poorly. In a word, in a small LR regime, stability cannot distinguish different minima at all.
> > > > >    - When the learning rate is large and batch size is small, our paper shows that the stable minima must be flat as measured by the Fro-norm of Hessian. However, the flat minima do not necessarily generalize well for general networks and data distributions.
> > > > >
> > > > > Therefore, in the large LR and small-batch regime, stability is useful in distinguishing flat minima and sharp minimal. What remains is to understand when and why flat minima generalize well. As mentioned previously,  in a separate work, we already proved that the Fro-norm of Hessian can bound the path norm of two-layer nets, thereby controlling the generalization gap. Therefore, **in such a case, one can conclude that all stable minima generalize well** as long as the LR and batch size of SGD  are independent of model size and sample size.  But we agree that stability cannot explain the generalization of SGD in a small LR regime, where SGD becomes close to gradient flow (GF).  (GF does find generalizable minima, though they generalize worse than SGD).
> > > > >
> > > > > &nbsp;
> > > > >
> > > > > **The role of SGD in the training process?**
> > > > > For simplicity, we compare GD and SGD from the same initialization. Assume that GD quickly converges to a sharp minimum that generalizes relatively bad. This suggests that there are many bad minima around the specific initialization. For SGD, the noise-induced stability prevents SGD from entering the basin of these bad minima, thereby enforcing SGD to continue to explore in flat regions.  It is currently unclear what kind of forces guide SGD to travel to flatter regions with better generalization properties. However, it is clear that it is the the noise-induced stability  that provides a possibility for SGD; otherwise, SGD will get stuck in bad minima.
> > > > >
> > > > > ---
> > > > >
> > > > > Lastly, we would like to thank you again for the comment, which helps make our work more solid, . We will particularly emphasize in the revision that the stability analysis is more relevant in a large LR and small-batch regime.

---

### Official Review · Reviewer_qyGZ · 2022-07-11

**Rating:** 7
**Confidence:** 3
**Soundness:** 3 good
**Presentation:** 3 good
**Contribution:** 3 good

**Summary:**

The paper provides an explanation of the phenomenon of SGD selecting flat minima by relating the noise structure of SGD to its linear stability. In over-parameterized models with the square loss, it shows, exploiting the geometry awareness of SGD noise (that is provable for linear networks and RFMs), that the Hessian of an accessible minimum for SGD is bounded by a term depending only on the batch size an the learning rate.

**Questions:**

- figure 3: are the results robust w.r.t. other flatness notions?
- figure 5, it could be useful to add VGG11 to the right panel
- the paragraph "notion of flatness" at l110 may be made more clear. Also the role of the norm in the flatness notions is not clear. also, size-independence plays an important role

Minors:
- l96: initiated studied

**Limitations:**

The role of overparameterization and the specific loss type (quadratic) are not discussed

potential negative societal impact are not discussed

**Strengths And Weaknesses:**

Strenghts:
- the topic of study is relevant
- the quantitative analysis is extensive, going from simple models s.t. RFMs and linear models to modern deep neural networks
- the paper is well written and clear

---

> ### Author Response · Authors · 2022-08-02
> **The importance of size independence of flatness**
>
> A short explanation: The size-independence of flatness is crucial for ensuring the complexity of SGD solutions to be bounded independently of the model size, thereby ensuring the generalization in the over-parameterized regime. A detailed explanation is given below.
> 1. First, if we view the flatness, which is Hessian's fro-norm in the current paper, as a complexity measure that can effectively control the model's capacity, then the size independence implies that the complexity of SGD solutions does not increase as increasing the model size. Therefore, it is crucial for arguing that SGD finds generalizable solutions, in particular in the over-parameterized regime. Specifically, in a separate work (we will appropriately cite it in the final version of this submission), we already proved that for two-layer neural nets (under certain conditions),
>
> $$
>   \qquad\qquad\qquad \qquad gen-gap (\theta^*) \leq O\left(\frac{||H(\theta^*)||_F}{\sqrt{n}}\right) 	\qquad \qquad \qquad \qquad \qquad (P),
> $$
>
> where the hidden constant only depends on the input dimension linearly. Together with the size independence of $||H(\theta^*)||_F$ guaranteed by the linear stability,  we can conclude that SGD finds generalizable solutions, thereby explaining the **implicit Regularization**.
>
> 2. Secondly, this size independence of Hessian's Fro-norm is also a major difference between SGD and GD. For GD, the stability only guarantees that $\lambda_\max(H(\theta^*))\leq 2/\eta$. Therefore, in general, GD finds minima, where only the curvature of the sharpest direction is controlled. By contrast, SGD tends to find minima, where the landscape is uniformly flat in different directions due to that the whole spectrum of Hessian is controlled. A naive bound of the Hessian's Fro-norm of GD solution is
>
> $$
> \qquad\qquad\qquad ||H(\theta^*)||_F\leq \sqrt{\text{rank}(H(\theta^*))}  \lambda_\max (H(\theta^*))\leq \frac{2\sqrt{\min(n,p)}}{\eta},
>   $$
>
> where $n,p$ denotes the sample and model size, respectively. Plugging this into the preceding generalization bound (P) only yields a vacuous/trivial bound of generalization gap. This partially explains why SGD generalizes better than GD.
>
> In a word, the size independence of Hessian's Fro-norm is critical to explaining why SGD selects generalizable minima and distinguishes SGD from GD.

---

> ### Author Response · Authors · 2022-08-02
> **figure 3: are the results robust w.r.t. other flatness notions?**
>
> It really depends on which flatness notion is used. For example,  the largest eigenvalue of Hessian should keep nearly unchanged when increasing the model size, since stability ensures $\lambda_{\max}(H)\leq 2/\eta$.  However, stability cannot provide direct control of the trace of Hessian and it may depend on model size more significantly. The following tables compare how the Fro-norm and trace of Hessian change with model size for linear networks and fully-connected networks. Each cell reports both the average and standard deviation (the value in parenthesis) of trace and fro-norm over 10 independent runs.
>
> Fully-connected networks.
>
> |network width/fully-connected nets|Fro-norm|Trace|
> |---|---|---|
> |10|3.4 (1.2)|6.0 (2.5)|
> |20|2.5 (0.3)|5.0 (0.9)|
> |40|2.9 (0.2)|7.4 (1.4)|
> |80|3.1 (0.1)|9.7 (1.6)|
> |160|3.1 (0.1)|12.6 (1.3)|
> |320|3.5 (0.1)|16.1 (1.0)|
>
> Linear networks.
> |network width/linear net|Fro-norm|Trace|
> |---|---|---|
> |10|2.0 (0.1)|5.3 (0.4)|
> |20|2.1(0.1)|6.0 (0.4)|
> |40|2.3 (0.1)|7.7 (0.5)|
> |80|2.4 (0.1)|9.5 (0.5)|
> |160|2.4 (0.1)|10.7 (0.2)|
>
> We see that as expected,  the trace indeed increases much more significantly than the Fro-norm. However, it is worth noting that the trace itself does not increase too much as well, which cannot be explained by our stability argument. This is probably attributed to the particularity of neural network models.  A plausible explanation is that: SGD tends to find minima for neural networks, where the Hessian is low rank. In such a case, the trace is close to the Fro-norm and the latter is provably bounded independently of the model size.  But explaining why SGD tends to find low-rank minima is beyond the scope of this paper.

---

> ### Author Response · Authors · 2022-08-02
> **Other questions**
>
>
> - **figure 5, it could be useful to add VGG11 to the right panel**
>
> Thanks for the suggestion. We have added the VGG11 result to the right panel of Figure 5b. Please take a look at the revised submission.
>
> - **the paragraph "notion of flatness" at l110 may be made more clear. Also, the role of the norm in flatness notions is not clear.**
>
> Sorry for that.  We refer the reviewer to lines 111-119 (marked in blue) of the revised submission, where the paragraph is rewritten to contain more details to make the statement more clear.

---

### Author Response · Authors · 2022-08-02
**Thank you to all reviewers**

We thank all the reviewers for their encouraging and insightful comments. We greatly appreciate the time and effort you spent on our paper. We will incorporate these valuable suggestions into the final version of our paper.

---

### Meta-Review · Area_Chair_5FJV · 2022-08-30

**Recommendation:** Accept
**Confidence:** Certain

**Metareview:**

The paper investigates an important topic of why SGD converges to flat minima.
Overall the reviewers felt that this is a nicely written paper with a nice contribution to the state of the art.




**Award:**

No

---

### Decision · Program_Chairs · 2022-09-14

Accept